# Characterization of *PtAOS1* Promoter and Three Novel Interacting Proteins Responding to Drought in *Poncirus trifoliata*

**DOI:** 10.3390/ijms21134705

**Published:** 2020-07-01

**Authors:** Jiang Xiong, Lian Liu, Xiaochuan Ma, Feifei Li, Chaolan Tang, Zehang Li, Biwen Lü, Tie Zhou, Xuefei Lian, Yuanyuan Chang, Mengjing Tang, Shenxi Xie, Xiaopeng Lu

**Affiliations:** 1Department of Horticulture, College of Horticulture, Hunan Agricultural University, Changsha 410128, China; xiongjiang@stu.hunau.edu.cn (J.X.); durian@stu.hunau.edu.cn (L.L.); maxiaochuan@stu.hunau.edu.cn (X.M.); lifly@hunaas.cn (F.L.); tang126@stu.hunau.edu.cn (C.T.); lizehang@stu.hunau.edu.cn (Z.L.); LBW@stu.hunau.edu.cn (B.L.); 8899@stu.hunau.edu.cn (T.Z.); 1753891547@stu.hunau.edu.cn (X.L.); changyuanyuan@stu.hunau.edu.cn (Y.C.); mengjing@stu.hunau.edu.cn (M.T.); shenxixie@163.com (S.X.); 2National Centre for Citrus Improvement, Changsha 410128, China; 3Institute of Horticulture, Hunan Academy of Agricultural Science, Changsha 410125, China

**Keywords:** allene oxide synthase, promoter, *Poncirus trifoliata*, interaction

## Abstract

Jasmonic acid (JA) plays a crucial role in various biological processes including development, signal transduction and stress response. Allene oxide synthase (AOS) catalyzing (13S)-hydroperoxyoctadecatrienoic acid (13-HPOT) to an unstable allene oxide is involved in the first step of JA biosynthesis. Here, we isolated the PtAOS1 gene and its promoter from trifoliate orange (*Poncirus trifoliata*). PtAOS1 contains a putative chloroplast targeting sequence in N-terminal and shows relative to pistachio (*Pistacia vera*) AOS. A number of stress-, light- and hormone-related cis-elements were found in the *PtAOS1* promoter which may be responsible for the up-regulation of *PtAOS1* under drought and JA treatments. Transient expression in tobacco (*Nicotiana benthamiana*) demonstrated that the P_−532_ (−532 to +1) fragment conferring drive activity was a core region in the *PtAOS1* promoter. Using yeast one-hybrid, three novel proteins, PtDUF886, PtDUF1685 and PtRAP2.4, binding to P_−532_ were identified. The dual luciferase assay in tobacco illustrated that all three transcription factors could enhance *PtAOS1* promoter activity. Genes *PtDUF1685* and *PtRAP2.4* shared an expression pattern which was induced significantly by drought stress. These findings should be available evidence for trifoliate orange responding to drought through JA modulation.

## 1. Introduction

In the south of China, seasonal drought happening from July to September generally damages the citrus industry regularly. Trifoliate orange (*Poncirus trifoliata*) is the main citrus rootstock in China because of its excellent tolerance or resistance to stresses including drought. However, its specific mechanism in drought tolerance is still not known well. Jasmonic acid (JA) is a kind of octadecanoid derivative synthesized from α-linolenic acid and released from galactolipids of chloroplast membranes. JA and its derivatives (JAs) serve as signaling molecules to regulate diverse aspects of plant life including leaf senescence [1], tuber formation [2], tendril coiling [3], filament elongation [4], biotic and abiotic responses [5,6,7].

JAs play positive roles in plants’ abiotic stress response. Previous studies demonstrated JA or MeJA could modulate plant physiological responses to abiotic stress. For instance, exogenous JA induced rapeseed seedlings’ higher tolerance to drought through improvements in fresh weight, chlorophyll content and relative water content [8]. Improvement in drought tolerance lead by exogenous MeJA was also found in wheat [9] and soybean [10]. On the other hand, stress was followed by endogenous JAs accumulation in most cases, which implied JA could act a positive regulator in stress tolerance [11]. Salt-tolerant tomatoes exhibited higher endogenous JA under whatever normal condition or salt treatment [12]. Salt-resistant mazes differed in JA content at the first phase of salt stress and that might work through Na^+^ uptake reduction but exclusion improvement in the roots [13]. Moreover, MeJA enhanced salt tolerance in wheat seedlings through decreases in H_2_O_2_ and MDA, and increments of antioxidant activities including POD, CAT and APX [14]. In a JA-deficient tomato, activity decreases in antioxidants accelerated oxidative stress by salt treatment, indicating that endogenous JA might play a positive role in plant salt tolerance [15].

Allene oxide synthase (AOS) converting (13S)-hydroperoxyoctadecatrienoic acid (13-HPOT) to an unstable allene oxide is involved in the first step of JA biosynthesis. AOS belongs to the superfamily of cytochrome P450. It is likely to be a key regulator in JA’s biosynthesis [16]. Over-expression of flax AOS in potato led to an increase in endogenous JA level [17]. After wounding, over-expression of *AtAOS* exhibited an elevated endogenous JA level 2- to 3-fold higher compared with the wild type [18]. Silencing of the *NaAOS* gene resulted in a JA content decrease in tobacco [19]. Defense gene activation was determined by *LeAOS* expression in tomato; exogenous MeJA treatment could not activate the expression of wound-inducible *PIN II* (proteinase inhibitor II) when *LeAOS* expression was inhibited [20]. Silencing *StAOS2* in potato significantly reduced endogenous JA content and compromised disease resistance [21]. Furthermore, AOS was also found acting a positive role in plant drought tolerance. Over-expression of cabbage *BaAOS* in Arabidopsis increased basal JA levels compared with wild type: with drought stress, transgenic lines exhibited a drastic increase in endogenous JA and higher tolerance [7].

Transcription factor interacts with cis-acting elements in a gene’s promoter region and activates transcription. In watermelon, lots of stress- and hormone-related cis-elements were found in the *ClAOS* promoter, indicating their possible roles in stress and hormone responses [22]. In Arabidopsis, intact plants showed weak *AtAOS1* promoter activity but that significantly increased at 3 h after mechanical wounding, suggesting the *AtAOS1* promoter could be an inducible dramatically by injury [23]. Promoter polymorphisms of *GmAOS1* associated significantly with soybean defense against common cut worm (*Spodoptera litura Fabricius*) attack [24].

A large number of AOS genes were cloned and characterized in diverse plants, however, there was still limited information about how it worked [21,25,26]. Our previous study suggested accumulations of JA and proline in citrus increased significantly by drought [27]. In this study, the coding sequence and promoter of *AOS1* in trifoliate orange (*PtAOS1*) were characterized. Gene expression and *PtAOS1* promoter activity were detected here. Afterward, a core region contributing to the promoter activity of *PtAOS1* was revealed through 5’ deletion assays. Using yeast one-hybrid and dual-fluorescence systems, three interacting proteins binding with the *PtAOS1* promoter were identified and named PtDUF886, PtDUF1685 and PtRAP2.4. Responses of *PtDUF886*, *PtDUF1685* and *PtRAP2*.4 to drought stress were investigated further.

## 2. Results

### 2.1. PtAOS1 Gene Expression and its Induction by Drought and JA Treatments

Full-length *PtAOS1* was isolated from trifoliate orange. Sequence analysis showed that *PtAOS1* encoded a protein of 499 amino acids with a predicted molecular mass of 59.5 kD. The PtAOS1 protein contains a chloroplast transit peptide in N-terminal that might function as a chloroplast localization signal. The phylogenetic tree showed that AOSs were divided into four groups in dicots (Figure 1). Except CsAOS of sweet orange (*Citrus sinensis*), PtAOS1 exhibited high similarity with pistachio (*Pistacia vera*) AOS. In addition, AOSs from walnuts (*Juglans regia*), bayberry (*Morella rubra*), California valley oak (*Quercus lobata*), cork oak (*Quercus suber*) and mulberry (*Morus notabilis*) were close to PtAOS1 in their evolutionary process.

Expression patterns of *PtAOS1* responding to drought stress and exogenous JA treatment were analyzed. The transcript of *PtAOS1* was induced significantly by exogenous JA within 12 h. Peak expression happened at 24 h after JA treatment, about 30-folds higher relative to the control. At 72 h after treatment, *PtAOS1* expression decreased sharply, still being about 3.2-folds higher relative to the control (Figure 2A). Expression of *PtAOS1* was induced soon after drought stress (Figure 2B). Up-regulation of *PtAOS1* was found immediately at 70% soil relative water content (RWC) in which citrus plant grows well generally. From 70% to 50% RWC, its expression level was up-regulated, about 2-folds higher than the control. From 40% to 20% RWC, the *PtAOS1* transcript decreased slightly.

### 2.2. Characterization of PtAOS1 Promoter

The promoter of *PtAOS1* containing 1181 bp upstream of the start codon was isolated and analyzed. The promoter fragment typically comprised an element involved in MeJA responsiveness (CGTCA-motif), an abscisic acid responsiveness element (ABRE) and an MYB binding site (MBS) mentioned drought stress (Figure 3). Additionally, cis-acting elements in defense and stress responsiveness (TC-rich repeats) and element essentials for anaerobic induction (ARE) were found also. An element conferring a high transcript level (5UTR Py-rich stretch) was found at −70 to −60 bp upstream of the start codon which might play a vital role in gene expression. A number of light responsive elements existed in the *PtAOS1* promoter such as AE-box, box4, BoxI, G-box, GA-motif, MNF1 and Sp1.

Transient expression assays were used to examine a core region in the *PtAOS1* promoter. Promoter P_−1181_ (−1181 to +1) and its truncated fragments P_−1062_ (−1062 to +1), P_−922_ (−922 to +1), P_−668_ (−668 to +1), P_−532_ (−532 to +1) and P_−265_ (−265 to +1) were assigned to drive a yellow fluorescent protein (YFP) in tobacco leaves. A YFP signal driven by P_−1182_ was detected successfully in tobacco epidermal cells which illustrated the transcription activation of the *PtAOS1* promoter. A YFP driven by P_−1062_, P_−922_, P_−668_ and P_−532_ fragments exposed a fluorescence signal equally with that using P_−1182_. However, a YFP signal with a P_−265_ fragment drive decreased sharply, implying a core function lost (Figure 4A–C).

To demonstrate the responses of the *PtAOS1* promoter to MeJA and abscisic acid (ABA), 50 µM MeJA and 25 µM ABA were applied on infiltrated tobacco leaves. Results showed YFP was stimulated significantly after MeJA and ABA treatments when most truncated promoters were used except only P_−265_ (Figure 5). No fluorescent signal was detected at all when a YFP was driven by P_−265_.

### 2.3. Proteins Recognizing PtAOS1 Promoter

The yeast one-hybrid screening system was employed to isolate proteins interacting with the *PtAOS1* promoter. Basing on the core region of detection, P_−532_ was selected for bait vector construction. The minimal inhibitory concentration of Aureobasidin A (AbA) for the bait yeast strain was 300 ng/mL. After a screen in a cDNA library prepared using drought-treated trifoliate orange leaves, three clones exhibited AbA resistance on selective plates (Figure 6A,B). Sequencing for three clones suggested they were *PtDUF886*, *PtDUF1685* and *PtRAP2.4*, respectively. One-to-one re-transformations for three proteins showed all of the three still grew normally on SD/-Leu medium with 300 ng/mL AbA (Figure 6C). To confirm the interactions, dual luciferase assays were performed additionally. Results suggested that *PtDUF886*, *PtDUF1685* and *PtRAP2*.4 could enhance P_−532_ drive activity 2.1-, 3.6- and 4.5-folds higher, respectively, relative to the control (Figure 6D).

### 2.4. Expression Profiles of Three Interacting Proteins under Drought Stress

Expression profiles of three proteins indicated they all responded to drought sensitively in trifoliate orange. Along with RWC decrease, the transcripts of *PtRAP2*.4 and *PtDUF1685* increased and peaked at 50% RWC, and then decreased gradually (Figure 7A,B). Gene *PtDUF886* exhibited 2-folds higher expression at 70% and 60% RWC. After a little decrease from 60% to 50% RWC, *PtDUF886* expressed stably from 50% to 20% RWC with a still higher expression than that without drought (Figure 7C).

## 3. Discussion

JAs were vital phytohormones participating in plant development and responding to biotic and abiotic stresses. Allene oxide synthase (AOS) catalyzes the first step in JA biosynthesis and the encoding gene works outstandingly after abiotic and biotic stresses [6,23]. Generally, plant drought tolerance correlated with endogenous JA accumulation. Expression of *CitAOS* in citrus plants increased gradually under drought stress along with JA accumulation [27]. In chickpea, a drought-tolerant variety exhibited earlier activation of *MtAOS* and higher JA content after drought treatment [28]. Transcriptome analysis revealed that *HaAOS* in sunflower root was up-regulated under salinity treatment [29]. Expression of *AtAOS* was rapidly induced at 15–30 min after wounding in Arabidopsis [30]. In tomato, insect chewing would lead to expression increases in both *LeAOS* and *LeHPL* [31,32]. Over-expression of AOS genes could improve plant tolerance to biotic and abiotic stress, and that probably works through an endogenous JA increase. Over-expression of the flax AOS gene in potato led to about 8- to 12-fold increase in endogenous JA [33]. With *CsAOS2* over-expression in tobacco, higher JA content was detected at 1 h after mechanical damage [34]. Transgenic rice with *OsAOS2* exhibited abundant *OsAOS2* transcripts and a higher endogenous JA level after pathogen infection [26]. Tobacco with over-expressed *TaAOS* exhibited higher chlorophyll content and displayed stronger tolerance to ZnCl_2_ stress [35]. In this study, expression of *PtAOS1* was significantly induced by drought in trifoliate orange (Figure 2B) showing a similar characteristic with that in other crops. In addition, *PtAOS1* expression was also dramatically up-regulated after JA treatment (Figure 2A) which was believed to enhance plant tolerance to drought. These results indicated that JA and *PtAOS1* worked really well in the drought tolerance of trifoliate orange.

It is widely known that a gene expression induced by an external stimulus was modulated by a regulatory motif in the promoter region [36,37,38]. Investigations of promoters help to understand the specificity mechanism of plant genes responding to abiotic stresses. For example, the *GmRD26* (a NAC transcription factor) promoter has been characterized as being involved in stress and ABA response; ABRE motifs and ABA responsive elements were thought to be the key factor influencing the stress-responsive gene expression regulated by *GmRD26* under drought stress [39]. In transgenic tobacco, disruption of an E-box/ABRE-like motif in the storage protein *napA* promoter led to the complete abolishment of reporter gene expression, suggesting that E-box is a core motif conferring *napA* promoter activity [40]. Here, the *PtAOS1* promoter contained cis-acting elements engaging in defense and stress responsiveness like an MYB binding site [41] and two phytohormone response elements, a CGTCA-motif and an ABRE element [42]. These implied an integrated way that trifoliate orange responded to drought through the JA pathway. Besides, the *PtAOS1* promoter was enriched in some light-responsive elements such as AE-box, G-box and MNF1 motif, indicating possible involvement in light response [43,44].

Identification of a core region in the promoter could be reached by promoter deletion. The grapevine vacuolar processing enzyme *VvβVPE* promoter was demonstrated as seed-specific, whereas this specificity vanished with a deletion from −1306 to −1045 bp [45]. In rice, sequence −1820 to −1525 bp harboring multiple core cis-acting elements including both CAAT-box and TATA-box was a core region of the *OsHAK1* promoter; further, the 5’ deletion analysis demonstrated that the promoter region −3037 to −1821 bp was indispensable for osmotic stress response [46]. In the present work, promoter deletion assays demonstrated that the region −532 to −265 bp was core-functional for the *PtAOS1* promoter (Figure 4C). Hormone treatments indicated the *PtAOS1* promoter responds to MeJA and ABA well, but the function lost all when the region −532 to −265 bp was absent. These evidences implied that the region −532 to −265 bp was indispensable for whatever *PtAOS1* promoter activity or MeJA and ABA responses. Further sequence analysis showed an ABRE element, a CGTCA-motif, three light-responsive elements and three CAAT-box were harbored in this promoter fragment.

Our previous reports suggested that the JA [27] and ABA [47] pathways were two of the causes of drought tolerance in trifoliate orange. CGTCA-motif had been previously identified as a binding site for transcription factors involved in JA signal transduction [48]. GUS activity showed that the prosystemin gene (*SlPS*) promoter region −221 to +40 bp was sufficient for JA-responsive transcription activation in tomato (*Solanum lycopersicum*) through a series of 5’ deletions [49]. Similarly, in kiwifruit, a CGTCA-motif in the ascorbic acid (AsA) synthesis gene *GGP* promoter was deduced to a function of regulating *GGP* gene expression when subjected to JA [50]. ABRE existing in −532 to −265 bp in the *PtAOS1* promoter was known as a major cis-acting regulatory element in ABA-dependent drought tolerance. An ABRE element was required for stress-responsive gene expression induced by osmotic stress in an ABA-dependent pathway. Gene *OSBZ8* was considered to regulate ABA-mediated transcription; the intensity of OSBZ8 binding to an ABRE element was high and constitutive in salt-tolerant rice cultivars compared with salt-sensitive cultivars [51]. Expressions of *AREB1*, *AREB2* and *ABF3* were induced by drought, ABA treatment and high-salinity in Arabidopsis: these three genes functioned as transcription activators in ABA-dependent response through directly binding to an ABRE sequence in the *DREB2A* promoter [52]. In this study, JA enhanced *PtAOS1* promoter activity but the inducibility was significantly reduced by a 268 bp deletion which contained a CGTCA-motif and an ABRE-element. These results demonstrated a CGTCA-motif and an ABRE-element located in −532 to −265 bp of *PtAOS1* promoter could be responsible for the JA drought response in trifoliate orange.

Limited information is known about AOS gene regulation. A recent report revealed a JAV1-JAZ8-WRKY51 (JJW) complex repressing JA biosynthesis through binding to the *AtAOS* promoter in healthy Arabidopsis. However, the JJW complex would disintegrate and activate JA biosynthesis once the plant was injured by mechanical damage or insect attack [53]. Similarly, OsDOF24 acted as a repressor for rice leaf senescence through binding directly to the *OsAOS1* promoter. A reduction in endogenous JA by over-expressed *OsDOF24* affected multiple physiological processes including leaf senescence, plant architecture and grain yield [54]. Here, we constructed and screened a cDNA library of trifoliate orange using the *PtAOS1* promoter bait, and three interacting proteins PtDUF886, PtDUF1685 and PtRAP2.4 were obtained (Figure 6A–C). The dual luciferase assay demonstrated all three transcription factors could enhance promoter activity, suggesting their positive roles in JA biosynthesis (Figure 6D). Domain of unknown function (DUF) proteins belong to a large uncharacterized protein family in the Pfam database [55]. The transcript level of *OsDUF810.7* was significantly induced by drought in rice: over-expression of *OsDUF810.7* in *E. coli* improved bacterial tolerance to salt and drought by enhancing antioxidant activities [56]. Expression levels of *OsDUF829.2* and *OsDUF829.4* in rice were significantly induced by salt and heat stress: over-expressions of them in *E.coli* improved its resistance to salt stress [57]. Expression of *DUF1618* genes differed in diverse rice cultivars and in responding to stress and hormone treatments [58]. In the present work, *PtDUF1685* and *PtDUF886* encoding two uncharacterized proteins, and transcripts of them were induced significantly by drought in trifoliate orange (Figure 7B,C). Our results indicated that *PtDUF1685* and *PtDUF886* probably functioned in drought tolerance through transcriptional regulation to *PtAOS1* in trifoliate orange. ERF transcription factors have been known to work in ethylene signal transduction and dehydration response [59,60,61,62]. Gene *GmERF3* expression was induced by abiotic stress (high-salinity and drought) and plant hormones (JA, SA, ET and ABA) in soybean; over-expression of *GmERF3* in tobacco improved its drought tolerance [63]. Similarly, over-expression of *TaERF3* in wheat also exhibited tolerance enhancement to salt and drought stresses [64]. Several ERF transcription factors were also found to be involved in JA signaling. Gene *ORCA3* encoding an AP2/ERF transcription factor was involved in JA signaling via directly binding to the jasmonate- and elicitor-responsive element of the Strictosidine synthase gene promoter [65]. ERF member *ORA59* acted as the key regulator of JA and ET responsiveness [66]. Genes *AaERF1* and *AaERF2* from *Artemisia annua* were both strongly induced by JA and acted as positive regulators of artemisinin biosynthesis [67]. Here, *PtRAP2*.4 with an AP2 domain was an AP2/ERF transcription factor and its expression was induced by drought significantly (Figure 7A). It could be concluded that *PtRAP2.4* played roles in not only the regulation of JA biosynthesis but also the crosstalk between the JA and ethylene signals. In this study, three proteins interacting with the *PtAOS1* promoter were identified and this probably offers new insights for JA responding to drought. These findings also help to understand more about drought tolerance of trifoliate orange and provide more information for citrus rootstock breeding. Drought stress would lead to multiple physiological and biochemical changes in diverse pathways. It would be essential to characterize these three novel proteins in regulating gene expressions, modulating JA synthesis and effecting an antioxidant system and so on. All the above contribute to a better understanding in the regulatory mechanism of trifoliate orange to drought stress.

In conclusion, we identified and analyzed *PtAOS1* from trifoliate orange which was induced by drought, ABA treatment and JA treatment. A region from −532 to −265 bp in the *PtAOS1* promoter was indispensable for promoter activity and ABA and JA responses. An ABRE element and a CGTCA-motif in the *PtAOS1* promoter were probably responsible for *PtAOS1* responding to drought. Further, three novel proteins which were drought-inducible and worked through binding to the *PtAOS1* promoter were identified. These findings may offer new insights into the drought tolerance of trifoliate orange through the JA pathway.

## 4. Materials and Methods

### 4.1. Plant Materials and Treatments

Six-month-old trifoliate orange (*Poncirus trifoliate*) seedlings in this assay were planted in a green house. All tobacco (*Nicotiana benthamiana*) seedlings were grown in a plant growth chamber with 16 h light/8 h dark cycle at 28 °C.

For drought treatment, trifoliate orange seedlings planted in a premix substrate were treated with no watering. Samples were collected when the relative water content of the substrate descended to 70%, 60%, 50%, 40%, 30% and 20%, respectively, detected by a soil moisture meter TDR 300 (Spectrum Technologies, Alpharetta, GA, USA). Control samples (CK) were collected from well-watered seedlings.

One mM MeJA was sprayed on the leaves of trifoliate orange. Samples were collected at 0, 12, 24 and 72 h after treatment. All samples were stored at −80 °C until use.

### 4.2. Cloning and Expression Pattern of PtAOS

The *PtAOS1* open reading frame (ORF) was PCR-amplified from trifoliate orange cDNA with primers (forward 5′-ATGGCATCCACTTCTCTATCTTT-3′, reverse 5′-AAAGCTTG CTCTCTTCAACGAC-3′) and constructed into a pM18-T vector (Takara, Dalian, China). The NCBI (http://www.ncbi.nl m.nih.gov/) database was used for a homologous sequence search through the BLAST program and protein sequences were aligned with the ClustalW 2.0 software. The phylogenetic tree for AOS proteins was constructed with the MEGA5.0 software using the neighbor-joining method. Chloroplast transit peptide was analyzed using TargetP (http://www.cb s.dtu.dk/services/TargetP) and ChloroP (http://www.cbs.dtu.dk/services/ChloroP/).

Gene expression patterns were assayed using mature leaves. Total RNA was isolated using RNAprep Pure Plant Kit (Tiangen, Beijing, China) according to the manufacturer’s protocol. The first strand of cDNA was synthesized from 1 µg of DNase treated total RNA using FastQuant RT Kit (Tiangen, Beijing, China) according to the manufacturer’s instruction. The primers used in the real-time quantitative PCR (qPCR) were as follows: *PtAOS1* (forward 5′-CCGTGTTCTGTCGTATCTTG-3′, reverse 5′-CCGTGTAGGTGGAGTGAA-3′), *PtDUF668* (forward 5′-CGAGCCA ATAGCCTATCAGT-3′, reverse 5′-GCTTCCACTCCAGATTATTCAC-3′), *PtDUF1685* (forward 5′-TGAGCATCAGAGCCAGAA-3′, reverse 5′-GATCGCCAGGACTAGAGAT-3′) and *PtRAP2.4* (forward 5′-AATGGCGGCTACAATGGATT-3′, reverse 5′-GGAGAAGCAGTTGGTGAAGT-3′). The qPCR was performed with SsoAdvancedTM SYBR^®^ Green Supermix (Bio-Rad, Hercules, CA, USA) using the CFX96 system (Bio-Rad, Hercules, CA, USA). The reaction mixture (total volume 10 µL) comprised 1 µL cDNA template, 0.5 µL primers (0.25 µM forward and 0.25 µM reverse), 5 µL 2× SYBR Green Supermix and 3.5 µL RNase-free water. The cycling parameters were 30 s at 95 °C, followed by 40 cycles of 95 °C for 5 s, 58 °C for 15 s. The relative expression level was normalized against the β-actin gene. Relative gene expression was calculated using the CFX96 software. Three biological replicates were performed for each sample.

### 4.3. Promoter Isolation and Vector Construction

Total genome DNA was extracted from trifoliate orange leaves using Plant Genomic DNA Extraction Kit (Takara, Dalian, China). The *PtAOS1* promoter was amplified using primers (forward 5′-CCTCTCCTCTTACTCTTCTC-3′, reverse 5′-TTCACAGCGATAATAATTGTTGT-3′) and cloned into a pMD18-T vector for sequencing. Core motif predicted by PlantCARE (http://bioinformatics.ps b.ugent.be/webtools/plantcare/html). The promoter of *PtAOS1* and truncated fragments (−1062 to +1, −922 to +1, −668 to +1, −532 to +1, −265 to +1) were PCR-amplified using a specific forward primer (F_−1062_ 5′-GAATTCAGACGACCAGCTTATGGCAAA-3′, F_−922_ 5′-GAATTCCTACTTGAGTTTGC GCCTT-3′, F_−668_ 5′-GAATTCCGTGGGCAATGCACTAGCCTA-3′, F_−532_ 5′-GAATTCAAATTGCAC CTATCAGCGTCA-3′, F_−265_ 5′-GAATTCTTGAAATGTTACGACTTGCGACT-3′) and the same reverse primer (5′-GGATCCTTCACAGCGATAATAATTGTTGT-3′). The PCR products were gel-purified and cloned into a modified pBI121-YFP vector with EcoRI and BamHI digestion. The corresponding vectors were named P_−1181_, P_−1062_, P_−922_, P_−688_, P_−532_ and P_−265_, respectively. All constructs were purified and transformed into agrobacterium (EHA105) for further transient expression.

### 4.4. Agrobacterium-Mediate Transient Expression in Tobacco

With *Nicotiana benthamiana*, the transient expression assay was performed as according to Sparkes [68] with minor modification. Strains preparation: streaking the corresponding agrobacterium strains on LB plates, incubate at 28 °C for 2 days. A single strain was selected and transferred into 3 mL liquid LB medium with 8 h shaking. This was followed by subsequent shaking in 50 mL LB until OD_600_ reached 0.6. Then, it was centrifuged at 3000 rpm and resuspended with infiltration buffer containing 10 mM 2-Morpholino ethanesulfonic acid hydrate (MES), 10 mM MgCl_2_ and 100 µM acetosyringone to an OD_600_ of 0.8, followed by incubating at 28 °C for 2 h. Infiltration: tobaccos leaves were injected from the back with a syringe (needle removed) until the water spots spread all over them. After 3 days, the YFP fluorescence intensity was detected with the LSM700 Fluorescence microscope (Zeiss, Oberkochen, Germany). Phytohormone treatment: tobacco leaves were treated with distilled water (CK), 50 µM MeJA or 25 µM abscisic acid (ABA) at 12 h after the infiltration, and the YFP signal was detected at 36 h after treatment.

### 4.5. Construction and Screening of a cDNA Library

The P_−532_ fragment of the PtAOS promoter was cloned into a pAbAi vector with digestion by SacI and XhoI. The construct was then digested by SmaI and then transformed into yeast strain Y1HGold according to the Matchmaker™ Gold Yeast One-Hybrid System (Clontech, Mountain View, CA, USA) protocol and selected on dropout medium lacking uracil (SD/-Ura). A positive yeast strain was then picked and resuspended in 0.9% NaCl to an OD_600_ of 0.002. Then, SD/-Ura with Aureobasidin A (AbA) medium was used to determine the minimal AbA inhibitory concentration. The cDNA library construction was performed following the manufacturer’s instruction for the Matchmaker^TM^ Gold Yeast One-Hybrid Library Screening System (Clontech, Mountain View, CA, USA). First, total RNA was extracted from trifoliate orange leaves using the method described before, and then mRNA was enriched and purified. Then, a cDNA for library construction was generated through first-strand SMART cDNA synthesis and long-distance PCR for double-strand cDNA (ds cDNA). The ds cDNA was purified with CHROMA SPIN+TE-400 columns. A one-hybrid cDNA library (20 µL SMART-amplified ds cDNA, 6 µL pGADT7-Rec) screen was performed according to yeast maker transformation system two. Library transformation reactions were spread on SD/-Leu with 300 mg/mL AbA with incubation at 30 °C for 3–5 days. Positive interaction clones were rescued and prey plasmids were isolated from selective plates. To quickly identify the clones, the T7 primer was used for sequencing, and plasmids were retransformed into bait yeast to confirm the interaction.

### 4.6. Dual Luciferase Assay

Full-length genes of *PtDUF886*, *PtDUF1685* and *PtRAP2.4* were amplified from trifoliate orange cDNA with gene-specific primers and cloned into the effector vector pGreenII 62-SK. The clone primers used were as follows: *PtDUF886* (forward 5’-GGATCCATGACATTTTCTTCTCCAATTCTC-3’, reverse 5’-GGATCCCCAATTGAGTTTCACCAGCTTCCAC-3’), *PtDUF1685* (forward 5’-GGATCCATGGGACGCTTTTCTTGTGA-3’, reverse 5’-GGATCCGCTGCACAACCTAACAG-3’) and *PtRAP2.4* (forward 5’-GGATCCATGGCGGCTACAATGGATTTC-3’, reverse 5’-GGATCCAGATAATATTGAAGCCCAATCA-3’). The P_−532_ fragment of the *PtAOS1* promoter was amplified and ligated into the reporter vector pGreenII 0800-LUC. The corresponding constructs were transformed into GV3101 component cells. Effector and reporter strains were separately re-cultivated on an LB plate with 50 mg/L kanamycin at 28 °C for 48 h. We resuspended the colony in MES buffer to an OD_600_ of 0.8 and incubated it for 2 h at 28 °C. Resuspended effector: reporter = 5:1 before infiltration was mixed up. After infiltration, tobacco seedings were moved back to the growth chamber with no watering for 3 days. Activity assays for firefly luciferase (LUC) and renilla luciferase (REN) were performed via the Dual-luciferase Reporter Assay System (Promega, Madison, WI, USA) according to the manufacturer’s instruction.

### 4.7. Statistical Analysis

All data were presented as mean ± SE with three independent replicates at least. The data were analyzed using the SPSS13.0 software running ANOVA according to Tukey’s multiple range test taking *p* < 0.05 as statistically significant.

## Figures and Tables

**Figure 1 ijms-21-04705-f001:**
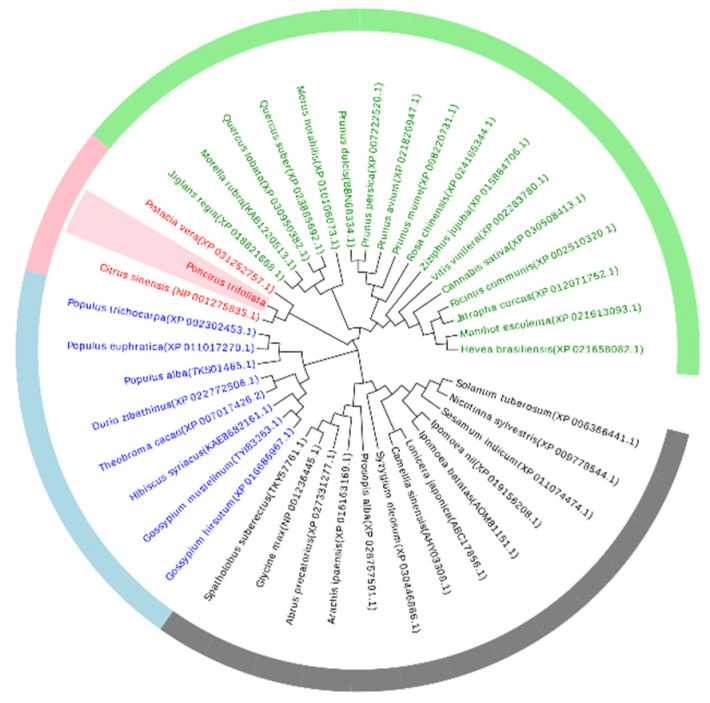
Phylogenetic analysis of allene oxide synthase (AOS) proteins among trifoliate orange and other plant species. The phylogenetic tree was constructed by the neighbor-joining method with 1000 replicates of bootstrap values.

**Figure 2 ijms-21-04705-f002:**
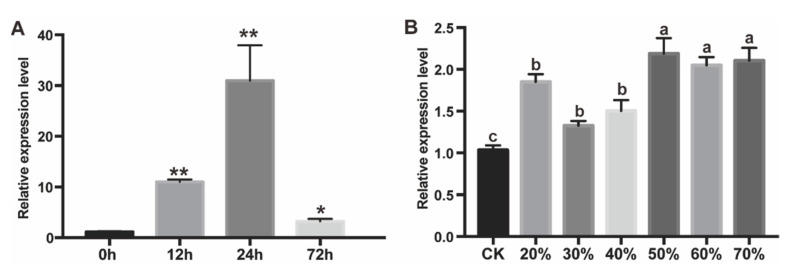
Expression patterns of *PtAOS1* in response to jasmonic acid (JA) (**A**); and drought (**B**). Data are shown as mean ± SD of three replicates. CK, control samples under well-watered condition. Asterisks indicate significant differences between treatment and control (*, *p* < 0.05 and **, *p* < 0.01). Different letters above the columns indicate significant differences at *p* < 0.05.

**Figure 3 ijms-21-04705-f003:**
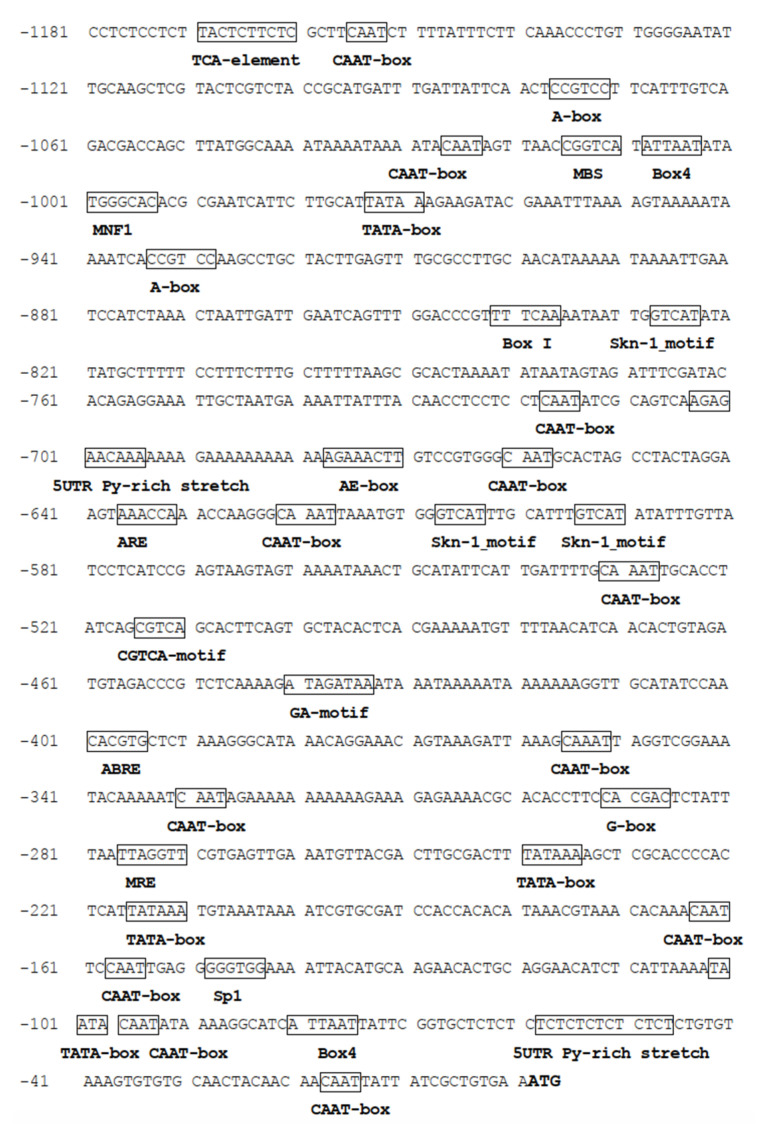
Core elements in the *PtAOS1* promoter. The ATG in bold is the start codon of the *PtAOS1* gene; the boxed region represents predicted cis-acting elements. 5UTR Py-rich stretch: cis-acting element conferring high transcription levels; A-box: cis-acting regulatory element associated with P- and L-box involved in induced transcriptional activity; ABRE: cis-acting element involved in the abscisic acid responsiveness; AE-box: part of a module for light response; ARE: cis-acting regulatory element essential for the anaerobic induction; Box4: part of a conserved DNA module involved in light responsiveness; Box I: light responsive element; CAAT-box: common cis-acting element in promoter and enhancer regions; CGTCA-motif: cis-acting regulatory element involved in MeJA responsiveness; G-box: cis-acting regulatory element involved in light responsiveness; GA-motif: part of a light responsive element; GT1-motif: light responsive element; MBS: MYB binding Site; MNF1: light responsive element; MRE: MYB binding site involved in light responsiveness; Skn-1_motif: cis-acting regulatory element required for endosperm expression; Sp1: light responsive element; TATA-box: core promoter element around −30 of transcription start; TC-rich repeats: cis-acting element involved in defense and stress responsiveness; TCA-element: cis-acting element involved in salicylic acid responsiveness.

**Figure 4 ijms-21-04705-f004:**
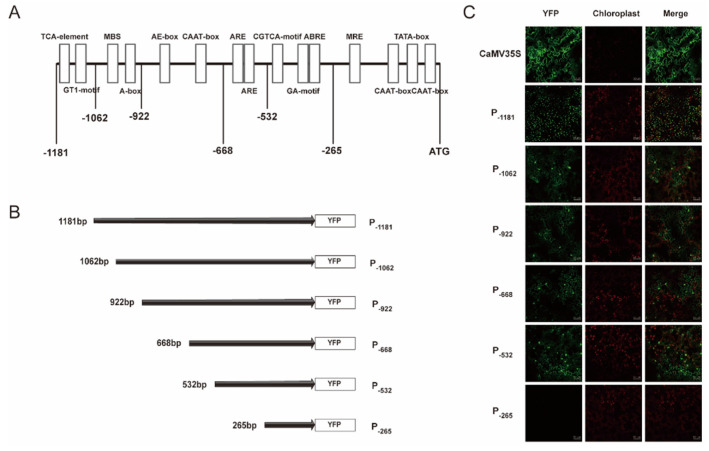
Construction for the *PtAOS1* promoter and transient expression in tobacco leaves. (**A**) Schematic diagram for the *PtAOS1* promoter. Boxes indicate cis-acting element in the promoter; (**B**) schematic diagrams of the truncated promoter construct; (**C**) transient expression of yellow fluorescent protein (YFP) driven by truncated promoter fragments in tobacco leaves.

**Figure 5 ijms-21-04705-f005:**
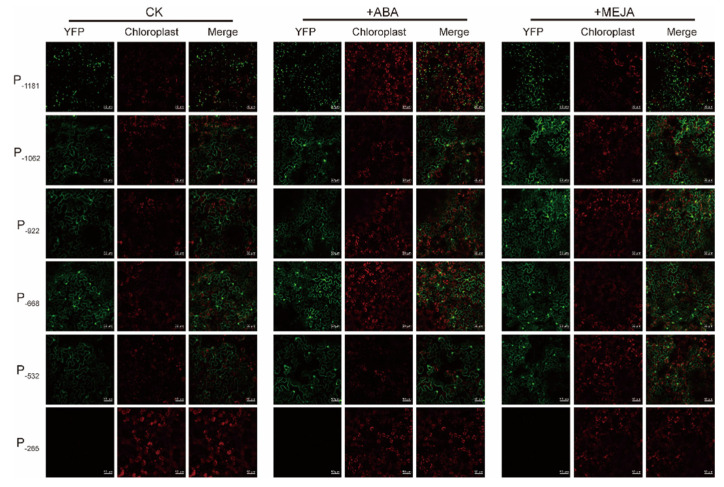
Responses of truncated *PtAOS1* promoters to ABA (abscisic acid) and JA (CK, control samples treated with distilled water). Scale bar in white represents 50 µm.

**Figure 6 ijms-21-04705-f006:**
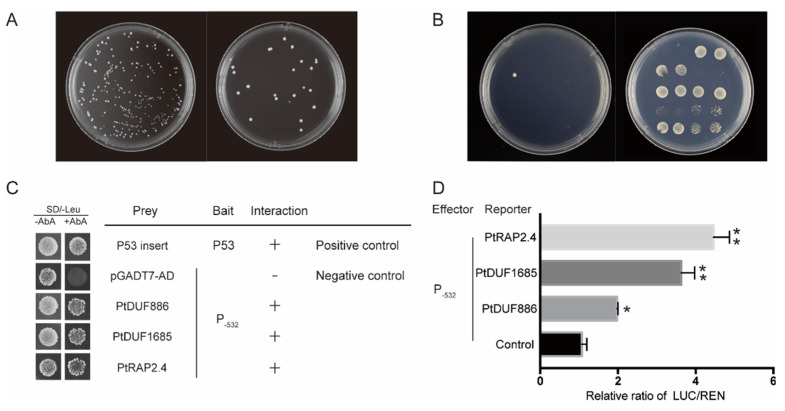
Isolation and identification of proteins interacting with the *PtAOS1* promoter. (**A**) Yeast library transformation and capacity identification; (**B**) screening and re-selection of positive clones on SD/-Leu with 350 ng/mL AbA; (**C**) growth of yeast strains co-transformed with prey and bait, negative control (bait/pGADT7) and positive control (p53/pGADT7-53) on selective medium; (**D**) dual luciferase assay by transient expression in tobacco. Asterisks indicate significant differences between drought treatment and control (*, *p* < 0.05 and **, *p* < 0.01).

**Figure 7 ijms-21-04705-f007:**
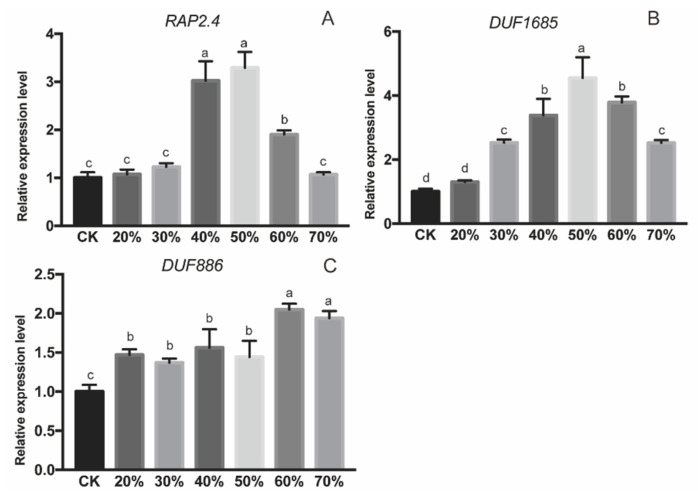
Expression pattern of *PtRAP2.4* (**A**), *PtDUF1685* (**B**) and *PtDUF886* (**C**) under drought stress. Different letters above the columns indicate significant differences at *p* < 0.05. CK, control samples under well-watered condition.

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
