# Peer review of "Characterization of PtAOS1 Promoter and Three Novel Interacting Proteins Responding to Drought in Poncirus trifoliata"

_ijms, 2020, doi:10.3390/ijms21134705_

Round 1

Reviewer 1 Report

The topic is interesting and the overall quality good. However, there are some changes required. First of all, in the title there should be mentioned the plant species. Furthermore, in the abstract, authors should also mention Nicotiana tabacum and the relative experiments. Why the species N. benthamiana is also mentioned in methods and materials? Is that correct? In figures, what is CK? Abbreviations should be explained. Also, the percentages in the x axis should be in a revers order (i.e. CK, 20%, 30% etc). In addition, the novelty of the findings and their potential use along with directions for further research should be also discussed.

Author Response

Thank you very much for remarks.
We have made appropriate additions to the manuscript now. 
All corrections and additions in the text are marked in yellow.

Point 1: First of all, in the title there should be mentioned the plant species.

Response 1: We have added the plant species in the title.

Point 2: Furthermore, in the abstract, authors should also mention Nicotiana tabacum and the relative experiments. Why the species N.benthamiana is also mentioned in methods and materials? Is that correct?

Response 2: We have made appropriate additions of relative experiments in the abstract. The tobacco species we used in this experiment was N.benthamiana, we are very sorry for our incorrect writing. We have made corrections in the full manuscript.

Point 3: In figures, what is CK? Abbreviations should be explained.

Response 3: We have provided the explanation of CK (control, treat method) in figure captions (Figure 2,5,7). And it was also added in method and materials.

Point 4: Also, the percentages in the x axis should be in a revers order (i.e. CK, 20%, 30% etc).

Response 4: According to reviewer's suggestion, we have reset the order of percentages in x axis as shown in Figure 2 and 7.

Point 5: In addition, the novelty of the findings and their potential use along with directions for further research should be also discussed.

Response 5: We have made some additions to discuss the finding novelty and further research work. All words were marked in yellow in the manuscript.

Reviewer 2 Report

The paper is very well done I found only minor typo to be set.

Row 84 Morella rubra should be in italic

Figure do not need a fullstop, please change all "Figure." to "Figure"

row 205 change "[42]" into "[42]."

Please rephrase the sentence in rows 210-213

Author Response

Thank you very much for your good comments.
All corrections and additions in the text are marked in yellow.

Point 1: Row 84 Morella rubra should be in italic.

Response 1: It was changed according to the reviewer's comments.

Point 2: Figure do not need a fullstop, please change all "Figure." to "Figure".

Response 2: We are very sorry for our incorrect writings. We have made correction according to the reviewer's comments.

Point 3: row 205 change "[42]" into "[42]."

Response 3: We have modified it according to reviewer's comments in the manuscript.

Point 4: Please rephrase the sentence in rows 210-213.

Response 4: Rows 210-213, the statements of "Basing on promoter deletion in rice, sequence -1820 to -1525 bp harboring two CAAT-box and CAAT-box element was believed responsible for promoter activity and promoter region -3037 to -1821 bp was indispensable for osmotic stress" were re-written as "In rice, sequence -1820 to -1225 bp harboring multiple core cis-acting elements including both CAAT-box and TATA-box was  a core region of OsHAK1 promoter; and the 5' deletion analysis demonstrated that promoter region -3037 to -1821 bp was indispensable for osmotic stress response"
